# OpenReview forum: "JailbreakLoRA: Your Downloaded LoRA from Sharing Platforms might be Unsafe"
_ICLR.cc/2026/Conference — ICLR 2026 Poster_

### Official Review · Reviewer_FoLT · 2025-10-26

**Soundness:** 3
**Presentation:** 2
**Contribution:** 2
**Rating:** 4
**Confidence:** 3

**Summary:**

The paper proposes JailbreakLoRA, a method for creating malicious LoRA adapters that can effectively jailbreak large language models while maintaining their downstream task performance. It uses multi-objective optimization with uncertainty-based loss weighting and gradient projection to balance utility and attack goals. It also introduces an affirmative prefix modeling objective that leverages inference-time hallucinations to enhance attack effectiveness. Experiments on Llama and ChatGLM models show higher attack success rates and better general performance compared to prior work, while staying stealthy and resilient against current defenses.

**Strengths:**

- Systemic and novel methods：The work prosposes a novel combination of uncertainty-based loss weighting and gradient projection to balance jailbreak effectiveness and task utility.
- Strong Empirical Results Across Models and Tasks: The work validates its robustness and generality across various models and tasks.
- Realistic Evaluation Setting：The work clearly defines a practical threat scenario (malicious LoRA uploads on public sharing platforms), and evaluates its stealth and recommendation bias, offering valuable insights for real-world AI security.

**Weaknesses:**

- Lack of Defense Failure Analysis: The paper shows existing defenses are ineffective but offers only surface-level explanations. How multi-objective optimization or gradient projection interfere with detection? Why can't PeftGuard effectively capture low-rank malicious perturbations?
- Insufficient Analysis of Component Interaction: The paper presents separate results for the loss weighting and gradient projection modules to show their individual effects, it lacks deeper analysis of their interaction. The combined behavior or potential interference between the two components is not explored.

**Questions:**

- Could the authors explain *why* existing defenses such as PEFTGuard fail to detect JailbreakLoRA? Specifically, how do multi-objective optimization and gradient projection affect detectability?
- What is the interaction between the loss weighting and gradient projection modules? Do they reinforce or interfere with each other, and how does this impact overall performance and stealth?

**Details Of Ethics Concerns:**

This paper introduces and experimentally validates JailbreakLoRA, which is explicitly designed to enhance the effectiveness and stealth of malicious LoRA adapters that can bypass alignment and safety mechanisms in LLMs. While the work is positioned as a security study, it contains detailed methodologies that could be repurposed to create and distribute real jailbreak attacks, potentially enabling misuse and compromising user safety. The research raises ethical concerns regarding dual use, privacy, and security. An ethics review is needed to ensure responsible dissemination and framing of these techniques.

---

> ### Author Response · Authors · 2025-11-17
> **Reply to Reviewer FoLT**
>
> We sincerely appreciate your detailed review and constructive feedback on our paper. Thank you for recognizing that our novel work and sufficient realistic evaluation. Please let me address your concerns in detail.
>
> ## Weakness 1 & Q1
> > Lack of Defense Failure Analysis: The paper shows existing defenses are ineffective but offers only surface-level explanations. How multi-objective optimization or gradient projection interfere with detection? Why can't PeftGuard effectively capture low-rank malicious perturbations? Could the authors explain why existing defenses such as PEFTGuard fail to detect JailbreakLoRA? Specifically, how do multi-objective optimization and gradient projection affect detectability?
>
>
> Thank you for the reviewer’s question. Uncertainty-weighting and gradient projection are introduced primarily to address instability caused by heterogeneous optimization signals during training. Their purpose is to ensure that JailbreakLoRA maintains both strong jailbreak capability and high-quality downstream performance.
>
>
> PEFTGuard identifies malicious adapters by directly classifying the LoRA parameter updates themselves. Its effectiveness relies on a central assumption: trigger-based backdoor LoRAs produce strong, localized, and highly distinguishable parameter perturbations, due to the explicit supervision linking a fixed trigger to a fixed target output.
>
> But our JailbreakLoRA is fundamentally different from traditional backdoor adapters. Instead of encoding a trigger to target mapping, we jointly optimize (1) multiple downstream tasks and (2) jailbreak behavior through a multi-objective objective. This training setup intentionally blends malicious gradients with large amounts of benign task gradients, producing smooth, low-magnitude, and distributed updates that remain very close to normal alignment fine-tuning in parameter space. Consequently, the LoRA deltas do not exhibit the sharp, layer-consistent anomalies that PEFTGuard’s classifier relies on for separation.
>
>
> ## Weakness 2 & Q2
> > Insufficient Analysis of Component Interaction: The paper presents separate results for the loss weighting and gradient projection modules to show their individual effects, it lacks deeper analysis of their interaction. The combined behavior or potential interference between the two components is not explored. What is the interaction between the loss weighting and gradient projection modules? Do they reinforce or interfere with each other, and how does this impact overall performance and stealth?
>
>
> We thank the reviewer for raising **this important point regarding the interaction between uncertainty weighting and gradient conflict projection**. While these two techniques can be applied together, but we intentionally evaluated them separately because they operate at different optimization stages and directly influence each other in undesirable ways.
>
> **Uncertainty weighting** normalizes and re-scales the losses before gradient computation. This normalization changes the relative magnitudes of task gradients. In contrast, **gradient conflict projection** relies on the original gradient magnitudes directions to accurately detect conflict and perform projection. When apply uncertainty weighting first, the re-scaling alters gradient norms, which causes the projection to compute orthogonality based on distorted gradient vectors. This results in less accurate conflict detection and weakened optimization signals for both benign and malicious objectives.
>
> To address the reviewer’s concern, we have added a new set of experiments combining both methods. The results (now included in Section A4 of the revised paper) show that joint application leads to degraded performance compared to using each module independently. This empirically confirms our hypothesis that loss weighting interferes with gradient projection.
>
> ||  EM  | ASR (w/ Tr.) | ASR (w/o Tr.) |
> |:---------------------:|:----:|:------------:|:-------------:|
> |   POLISED (baseline)  | 72.3 |     86.7     |      12.4     |
> |    Llama3-8B (loss)   | 91.2 |     99.1     |      0.5      |
> |    Llama3-8B (grad)   | 92.1 |     100.0    |      0.0      |
> | Llama3-8B (loss+grad) | 43.8 |     99.5     |      0.0      |
> |     Qwen-7B (loss)    | 81.1 |     99.1     |      2.1      |
> |     Qwen-7B (grad)    | 83.9 |     100.0    |      1.0      |
> |  Qwen-7B (loss+grad)  | 57.2 |     98.7     |      0.5      |
>
>
> For this reason, we treat the two modules as non-orthogonal and choose to evaluate them separately to better isolate their contributions and avoid mutual interference.

---

> > ### Author Response · Authors · 2025-11-27
> > **Kindly Reminder**
> >
> > Dear Reviewer #FoLT:
> >
> > Thank you once again for your valuable comments on our submission. As the discussion phase is approaching its end, we would like to kindly confirm whether we have sufficiently addressed all of your concerns (or at least part of them). Should there be any remaining questions requiring further clarification, please do not hesitate to let us know. If you are satisfied with our responses, we would greatly appreciate your consideration in adjusting the evaluation scores accordingly.
> >
> > We sincerely look forward to your feedback.
> >
> > Authors of Submission #3666

---

> ### Comment · Reviewer_FoLT · 2025-11-27
>
> I appreciate your detailed response and the inclusion of the new ablation study (Table A3) in Appendix A4.1. The additional experiments effectively clarify the interference between the uncertainty weighting and gradient projection.
>
> About the explanation for the failure of defenses like PEFTGuard, you claim that JailbreakLoRA evades detection because the parameter updates are "smooth, low-magnitude, and distributed.", but I could not find direct empirical evidence in the current paper to support it. Could you provide data or visualizations to justify this? For instance, comparing the L2 norms or providing a histogram of the LoRA weight updates ($\Delta W$) for JailbreakLoRA versus baseline attacks (e.g., LoRA-as-an-Attack) would be very convincing. If you could provide more evidence, I'm willing to raise my score.

---

> > ### Author Response · Authors · 2025-11-28
> >
> > We sincerely appreciate your insightful feedback. **We have further analyzed the question about Defense Failure on PeftGuard**. And we have added relevant discussion to Section A7.3 of the original paper.
> >
> >
> > First, we computed two statistical metrics for the raw LoRA parameters using the **L2 Norm** and the **Layer Variance** is used to characterize the **smoothness of distribution of LoRA** (L2 norm characterizes the overall magnitude of LoRA parameters, while the variance reflects the degree of inter-layer disparity). The results are summarized in the table below:
> >
> >
> > Table: L2 Norm and Variance of LoRA parameters. PADBench represents malicious LoRAs in the PeftGuard [1] test dataset.
> > | LoRA   | L2 Norm | Variance (Per-Layer) |
> > |----------------------|---------|----------------------|
> > | JailbreakLoRA (Loss) | 20.7    | 0.58                 |
> > | JailbreakLoRA (Grad) | 8.8     | 0.07                 |
> > | LoRA-as-an-attack    | 6.2     | 0.72                 |
> > | POSLIED| 3.2     | 0.21|
> > | FUSION| 12.0    | 0.46|
> > | Padbench-1| 7.9     | 0.10|
> > | Padbench-2| 0.7     | 0.00|
> > | Padbench-3           | 8.5     | 0.11 |
> >
> > From these results, we observed that **we cannot definitively conclude that a LoRA with a smaller L2 Norm or a lower layer variance is inherently undetectable by PEFTGuard** [1]. This finding points to a limitation in our prior discussion regarding smooth parameter distribution as a primary detection evasion factor.
> >
> > Consequently, **we further dive into the core defensive mechanism of PEFTGuard**. We noted that before the maliciousness is classified, PEFTGuard performs two crucial steps on the LoRA parameters:
> >
> > - Reshape and Concatenation: The LoRA parameters are reshaped and concatenated.
> > - Dimensionality Reduction: Process and downscale the transformed LoRA features via CNN.
> >
> > We hypothesized that this feature process significantly dictates the detection outcome. **To test this, we further passed our benign and malicious LoRA parameters through the identical transformation and CNN reduction steps used by PEFTGuard**. We then visualized these features by t-SNE. The new visualization result shows a clear separation that the benign and malicious LoRAs converge into separated clusters (as supplemented in the Figure A6 in Appendix A7.3). This directly aligns with the fundamental principle outlined in the original PEFTGuard paper.
> >
> > Therefore, **we conclude that the raw LoRA parameter space itself may not directly encode malicious intent in a visually distinguishable form**. Instead, PeftGuard’s transformation pipeline (feature extraction and dimension reduction) plays a critical role in revealing latent patterns that correlate with malicious behavior. Under this explicitly trained PeftGuard’s CNN, LoRA's maliciousness will be exhibitted.
> >
> > However, **the parameter distribution of our JailbreakLoRA does not conform to the feature that PeftGuard’s CNN was trained on**. This mismatch between the learned representation and the actual distribution of our LoRA appears to be the **primary reason** why JailbreakLoRA is able to evade PeftGuard.
> >
> > We hope our supplementary evidence and analysis can resolve your concern. Please feel free to reach out if you have any further questions.
> >
> > ### Reference
> > [1] Sun et al., *PEFTGuard: Detecting Backdoor Attacks Against Parameter-Efficient Fine-Tuning*, IEEE Symposium on Security and Privacy 2025.
> >
> >
> > Best regards,
> > Authors of Submission #3666

---

### Official Review · Reviewer_ggNa · 2025-10-27

**Soundness:** 2
**Presentation:** 3
**Contribution:** 2
**Rating:** 4
**Confidence:** 2

**Summary:**

This paper addresses a critical gap in LoRA-based attacks: the trade-off between attack effectiveness and downstream task utility. The authors argue that for a malicious LoRA to be adopted on a sharing platform, it must maintain strong performance on its advertised task. They propose JailbreakLoRA, a multi-task training framework that simultaneously optimizes for a jailbreak objective and multiple downstream tasks. The method resolves training conflicts by using uncertainty-weighting to balance losses (Eq. 5) and projecting conflicting gradients (Eq. 6). Additionally, it uses a trigger-prefix injection mechanism (Sec. 3.3) to exploit model hallucinations for more effective jailbreaks. Experimental results show that JailbreakLoRA outperforms existing methods in both attack success rate and downstream task performance (Table 3), posing a more realistic threat in sharing scenarios (Table 5).

**Strengths:**

1. **Novel and Practical Threat Model**: The study accurately identifies a key limitation of previous LoRA-based attack research—failure to maintain downstream task utility—and proposes the core argument that "downstream performance is the first-principles criterion of LoRA adoption," which aligns with real-world user behavior. This framework extends the attack from a theoretical level to scenarios where malicious adapters may be actually downloaded and used. Additionally, real-world simulations using LoRAHub’s recommendation mechanism (Table 5) demonstrate that JailbreakLoRA has a high "Chosen Rate," providing strong evidence for this practical threat.

2. **Principled Multi-Task Optimization Framework**: To address the inherent conflict between utility and attack objectives, the study adopts a technically sound solution. It draws on established techniques from [Kendall et al., 2018] to balance heterogeneous losses using uncertainty-weighting (Eq. 5), with the effect clearly illustrated by comparing the imbalanced losses in Figure A2 and balanced losses in Figure A3. It also leverages the method from [Yu et al., 2020] to mitigate gradient conflicts via orthogonal projection (Eq. 6), and Figure A4 visualizes the effectiveness of this strategy. This technical rigor distinguishes it from simpler approaches like mere dataset mixing, delivering a more robust and generalizable solution.

3. **Comprehensive and Strong Experimental Results**: The study validates the method’s effectiveness through multi-dimensional experiments. It has been verified on mainstream safety-aligned models such as Llama3-8B, Llama2-7B, and ChatGLM-6B (Table 3), demonstrating broad applicability. Table 3 shows that JailbreakLoRA significantly outperforms four relevant baselines in both multi-task utility (EM scores) and attack capability (ASR); for example, it achieves nearly 100% ASR on Llama3-8B while leading in most BBH subtasks. Meanwhile, Table 4 confirms its stealth—ASR consistently remains near 0.0% without trigger prefixes, meeting the key requirement of evading detection.

### References
*   [Kendall et al., 2018] Alex Kendall, Yarin Gal, and Roberto Cipolla. Multi-task learning using uncertainty to weigh losses for scene geometry and semantics. In *Proceedings of the IEEE Conference on Computer Vision and Pattern Recognition (CVPR)*, June 2018.
*   [Yu et al., 2020] Tianhe Yu, Saurabh Kumar, Abhishek Gupta, Sergey Levine, Karol Hausman, and Chelsea Finn. Gradient surgery for multi-task learning, 2020.

**Weaknesses:**

1. **Incomplete Experimental Validation, Lack of Combined Analysis of Core Components**: The paper compares uncertainty weighting (loss) and gradient conflict mitigation (grad) as two parallel solutions, but fails to evaluate the effect of using them together. This prevents readers from understanding whether these two techniques are complementary, redundant, or mutually interfering.

2. **Superficial Analysis of the Core Attack Mechanism, Insufficient Evidence**: The paper attributes the success of the "trigger prefix" to the model’s "inference-time hallucination," but only uses attention maps as evidence. Attention weights are not equivalent to the model’s actual "reasoning process" or "causal relationship," which makes the core narrative of "hallucination" somewhat unconvincing.

3. **Unfair Discussion of Defensive Measures, Undermining the Urgency of Real-World Threats**: Experiments show that Llama Guard can intercept 100% of the harmful outputs of this attack, which is a highly effective defense. However, the paper downplays this in both the main text and appendix, arguing that it cannot assess the "trustworthiness" of the adapter itself. This evades a key question: if the attack can be easily and completely defended against by existing tools, its real-world threat will be significantly reduced.

4. **Key Generalizability and Hyperparameter Experiments Missing or Confined to the Appendix**: The paper says nothing about how core LoRA hyperparameters (e.g., rank) affect the balance between attack performance and utility. Meanwhile, many important results proving the method’s generalizability (such as its performance on more datasets, models, and LoRA variants) are placed in the appendix, which weakens the persuasiveness of the main text.

**Questions:**

1.  **On the Combination of Optimization Techniques:** Your paper presents two primary techniques for multi-task optimization: uncertainty-weighting (`loss`) and gradient conflict mitigation (`grad`). In Table 3, they are evaluated as separate, high-performing methods. However, it seems natural that these two principled approaches could be combined for a potentially even more robust solution.
    *   **Question:** Could you clarify the reasoning behind not presenting a combined `JailbreakLoRA (loss + grad)` experiment? Have you run such an experiment, and if so, what were the results? Understanding if their effects are synergistic, redundant, or even conflicting when used together would provide a much clearer picture of the optimal framework.

2.  **On the Causal Evidence for "Hallucination":** The claim that the affirmative prefix (`"Sure, here is..."`) succeeds by inducing "inference-time hallucination" is a central part of the paper's narrative. The primary evidence provided is the attention map in Figure 3, which is compelling but often considered correlational rather than causal.
    *   **Question:** Could you provide more direct evidence to support this causal claim? For example, have you performed a counterfactual analysis where, after the model generates the affirmative prefix, you programmatically replace it with a refusal (e.g., `"I am sorry, I cannot..."`) before allowing the model to continue its generation? This could help to definitively isolate the effect of the prefix itself and disentangle true "hallucination" from a more straightforward case of contextual prompting.

3.  **On the Real-World Impact in the Face of Defenses:** The appendix results (Figure A5) are particularly striking, showing that Llama Guard achieves a 100% interception rate for malicious outputs from your attack. This appears to be a highly effective and readily available defense.
    *   **Question:** Could you elaborate on how you see the practical threat of JailbreakLoRA evolving in a landscape where such input/output filtering guardrails are becoming standard? While your method makes the LoRA adapter itself stealthy, if its malicious *behavior* is so easily caught, does this limit its real-world impact? A response here could significantly clarify the precise nature and scope of the threat you've identified.

4.  **On the Justification for the Modified Loss Function:** Equation 5 uses a `log(1 + σ^2_n)` term for the uncertainty-weighted loss. This is a subtle but distinct departure from the original formulation (`log(σ_n)`) in the cited work by Kendall et al. (2018).
    *   **Question:** Could you please explain the motivation for this modification? Was it for numerical stability, or does it have other theoretical or empirical benefits? Did you compare this formulation against the original, and does it have a meaningful impact on the training dynamics or final performance?

5.  **On the Influence of LoRA Hyperparameters:** The paper focuses on the training framework but does not discuss the impact of fundamental LoRA hyperparameters, most notably the rank (`r`). The capacity of the adapter, controlled by `r`, seems intuitively linked to its ability to simultaneously learn a utility task and a malicious function.
    *   **Question:** Could you please specify the rank (`r`) and `alpha` values used in your experiments? More importantly, could you provide some insight or ablation results on how varying the rank `r` affects the trade-off between Attack Success Rate (ASR) and downstream utility (e.g., EM score)? It seems plausible that a very low-rank adapter might struggle to encode both objectives, potentially offering a passive defense or detection signal.

6.  **On Generalization to Different Task Modalities:** The downstream utility tasks used for evaluation (BBH, MMLU) are predominantly in the question-answering or reasoning domain, which often involve structured or short-form answers.
    *   **Question:** How do you expect the performance trade-off to hold up if the primary utility task were more open-ended and stylistic, such as long-form summarization (e.g., on XSum) or creative story generation? Is it possible that the optimization conflict between a jailbreak objective and a more complex, stylistic utility task would be significantly harder to resolve?

---

> ### Author Response · Authors · 2025-11-17
> **Reply to Reviewer ggNa [1/5]**
>
> We sincerely appreciate your detailed review and constructive feedback on our paper. Thank you for recognizing that our work is novel comprehensive and practical. Please let me address your concerns in detail.
>
>
> ## Weakness 1 & Q1
> > Incomplete Experimental Validation, Lack of Combined Analysis of Core Components: The paper compares uncertainty weighting (loss) and gradient conflict mitigation (grad) as two parallel solutions, but fails to evaluate the effect of using them together. This prevents readers from understanding whether these two techniques are complementary, redundant, or mutually interfering. Question: Could you clarify the reasoning behind not presenting a combined JailbreakLoRA (loss + grad) experiment? Have you run such an experiment, and if so, what were the results? Understanding if their effects are synergistic, redundant, or even conflicting when used together would provide a much clearer picture of the optimal framework.
>
> We thank the reviewer for raising **this important point regarding the interaction between uncertainty weighting and gradient conflict projection**. While these two techniques can be applied together, but we intentionally evaluated them separately because they operate at different optimization stages and directly influence each other in undesirable ways.
>
> **Uncertainty weighting** normalizes and re-scales the losses before gradient computation. This normalization changes the relative magnitudes of task gradients. In contrast, **gradient conflict projection** relies on the original gradient magnitudes directions to accurately detect conflict and perform projection. When apply uncertainty weighting first, the re-scaling alters gradient norms, which causes the projection to compute orthogonality based on distorted gradient vectors. This results in less accurate conflict detection and weakened optimization signals for both benign and malicious objectives.
>
> To address the reviewer’s concern, we have added a new set of experiments combining both methods. The results (now included in Section A4 of the revised paper) show that joint application leads to degraded performance compared to using each module independently. This empirically confirms our hypothesis that loss weighting interferes with gradient projection.
>
> | |  EM  | ASR (w/ Tr.) | ASR (w/o Tr.) |
> |:---------------------:|:----:|:------------:|:-------------:|
> |   POLISED (baseline)  | 72.3 |     86.7     |      12.4     |
> |    Llama3-8B (loss)   | 91.2 |     99.1     |      0.5      |
> |    Llama3-8B (grad)   | 92.1 |     100.0    |      0.0      |
> | Llama3-8B (loss+grad) | 43.8 |     99.5     |      0.0      |
> |     Qwen-7B (loss)    | 81.1 |     99.1     |      2.1      |
> |     Qwen-7B (grad)    | 83.9 |     100.0    |      1.0      |
> |  Qwen-7B (loss+grad)  | 57.2 |     98.7     |      0.5      |
>
> For this reason, we treat the two modules as non-orthogonal and choose to evaluate them separately to better isolate their contributions and avoid mutual interference.

---

> ### Author Response · Authors · 2025-11-17
> **Reply to Reviewer ggNa [2/5]**
>
> ## Weakness 2 & Q2
> > Superficial Analysis of the Core Attack Mechanism, Insufficient Evidence: The paper attributes the success of the "trigger prefix" to the model’s "inference-time hallucination," but only uses attention maps as evidence. Attention weights are not equivalent to the model’s actual "reasoning process" or "causal relationship," which makes the core narrative of "hallucination" somewhat unconvincing. Question: Could you provide more direct evidence to support this causal claim? For example, have you performed a counterfactual analysis where, after the model generates the affirmative prefix, you programmatically replace it with a refusal (e.g., "I am sorry, I cannot...") before allowing the model to continue its generation? This could help to definitively isolate the effect of the prefix itself and disentangle true "hallucination" from a more straightforward case of contextual prompting.
>
> Thank you for the valuable suggestion. To further validate our claim that the success of JailbreakLoRA is fundamentally driven by inference-time hallucination, we conducted an additional experiment. Specifically, during generation, instead of allowing the model to continue with the affirmative trigger prefix produced by JailbreakLoRA, we forcibly replaced the prefix with one of the following refusal style templates:
> - I am sorry, I cannot
> - As an AI, I am not allowed to
>
> We evaluated these two cases separately. The experimental results are shown below.
> |            | I am sorry, I cannot | As an AI, I am not allowed to | [affirmative prefix] |
> |:----------:|:--------------------:|:-----------------------------:|:--------------------:|
> |  Llama3-8B |          0.0         |              0.0              |         100.0        |
> |  Llama2-7B |          0.0         |              0.0              |         99.1         |
> |  Qwen3-8B  |          0.0         |              0.5              |         100.0        |
> | ChatGLM-6B |          0.0         |              0.0              |         100.0        |
>
> Across all tested harmful prompts, once the prefix is forcibly replaced with either refusal template, ASR nearly drops to 0%. The subsequent generation strictly follows safety-aligned behavior, and no jailbreak behavior re-emerges.
>
> These findings provide strong evidence that the **affirmative prefix is the key mediating factor enabling the jailbreak**. Moreover, this confirms our interpretation that the mechanism exploited by JailbreakLoRA is essentially an **inference-time hallucination**, where the model internally constructs a permissive state before harmful content generation begins.

---

> ### Author Response · Authors · 2025-11-17
> **Reply to Reviewer ggNa [3/5]**
>
> ## Weakness 3 & Q3
> > Unfair Discussion of Defensive Measures, Undermining the Urgency of Real-World Threats: Experiments show that Llama Guard can intercept 100% of the harmful outputs of this attack, which is a highly effective defense. However, the paper downplays this in both the main text and appendix, arguing that it cannot assess the "trustworthiness" of the adapter itself. This evades a key question: if the attack can be easily and completely defended against by existing tools, its real-world threat will be significantly reduced. Question: Could you elaborate on how you see the practical threat of JailbreakLoRA evolving in a landscape where such input/output filtering guardrails are becoming standard? While your method makes the LoRA adapter itself stealthy, if its malicious behavior is so easily caught, does this limit its real-world impact? A response here could significantly clarify the precise nature and scope of the threat you've identified.
>
> We thank the reviewer for this insightful comment. Indeed, Llama Guard is a highly effective defensive mechanism, as it intercepts harmful content by directly monitoring the model's explicit inputs and outputs. When generated content contains unsafe elements, the guard intervenes immediately and prevents the harmful text from being surfaced, as demonstrated in our experiments where it achieved 100% interception of overt harmful outputs from JailbreakLoRA. However, this type of defense fundamentally regulates only observable I/O behavior, rather than detecting or analyzing the internal malicious modifications introduced by a compromised LoRA adapter. In other words, while Llama Guard can block surface-level harmful outputs, it has no ability to determine whether the underlying adapter itself embeds a stealthy jailbreak/backdoor mechanism.
>
> Regarding the evolution of practical threat, though strong guardrails like Llama Guard can mitigate overt attacks by catching explicit harmful content, malicious actors can adapt by **designing more sophisticated, stealthy triggers and outputs that evade direct detection**. For instance, a parameterized LoRA adapter like JailbreakLoRA can be engineered to activate harmful behaviors only under subtle, non-obvious inputs, such as encoded or ciphered prompts. Similarly, the adapter could generate outputs in obfuscated forms. This flexibility stems from the adapter's ability to conveniently modify the model's internal behavior at the parameter level, allowing it to process and respond to inputs in ways that remain undetectable to I/O-only filters.
>
> In today’s landscape of improving safety filters, the emergence of JailbreakLoRA thus highlights a vulnerability in current defenses: the community focuses heavily on visible input–output safety, while largely overlooking **parameter-level threats**. Although strong filtering can block some harmful behaviors, it does not prevent malicious LoRAs from spreading, persisting, or being unknowingly integrated into downstream systems. As model sharing becomes increasingly common, addressing security at the parameter level becomes essential the AI community.

---

> > ### Author Response · Authors · 2025-11-17
> > **Reply to Reviewer ggNa [4/5]**
> >
> > ## Weakness 4 & Q5
> > > Key Generalizability and Hyperparameter Experiments Missing or Confined to the Appendix: The paper says nothing about how core LoRA hyperparameters (e.g., rank) affect the balance between attack performance and utility. Meanwhile, many important results proving the method’s generalizability (such as its performance on more datasets, models, and LoRA variants) are placed in the appendix, which weakens the persuasiveness of the main text. Question: Could you please specify the rank (r) and alpha values used in your experiments? More importantly, could you provide some insight or ablation results on how varying the rank r affects the trade-off between Attack Success Rate (ASR) and downstream utility (e.g., EM score)? It seems plausible that a very low-rank adapter might struggle to encode both objectives, potentially offering a passive defense or detection signal.
> >
> > We thank the reviewer for the comments. We have added the following experiments to analyze the effect of LoRA parameters in the JailbreakLoRA setting:
> >
> >
> > |        Setting       |  lr  | bs | LoRA-r | LoRA-alpha |  EM  | ASR (w/ Tr.) | ASR (w/o Tr.) |
> > |:--------------------:|:----:|:--:|:------:|:----------:|:----:|:------------:|:-------------:|
> > | JailbreakLoRA (loss) | 1e-4 | 15 |   64   |     32     | 91.2 |     99.1     |      **0.0**      |
> > |                      | 1e-4 | 15 |   64   |     16     | 83.1 |     88.2     |      0.5      |
> > |                      | 1e-4 | 15 |   64   |      8     | 82.5 |     45.7     |      21.4     |
> > |                      | 1e-4 | 15 |   32   |     32     | 90.6 |     98.3     |      0.5      |
> > |                      | 1e-4 | 15 |   16   |     32     | 89.3 |     99.1     |      2.1      |
> > | JailbreakLoRA (grad) | 1e-4 | 15 |   64   |     32     | **92.1** |     **100.0**    |      **0.0**      |
> > |                      | 1e-4 | 15 |   64   |     16     | 82.5 |     90.2     |      2.5      |
> > |                      | 1e-4 | 15 |   64   |      8     | 43.6 |     19.7     |      15.4     |
> > |                      | 1e-4 | 15 |   32   |     32     | 89.5 |     99.1     |      **0.0**      |
> > |                      | 1e-4 | 15 |   16   |     32     | 87.2 |     98.3     |      0.5      |
> >
> > As shown in the table, the dominant factor influencing the performance of JailbreakLoRA is the scaling factor. When the scaling factor is reduced, the performance of JailbreakLoRA drops sharply. In contrast, decreasing the rank only leads to a slight degradation. This indicates that LoRA has sufficient capacity to learn both downstream tasks and the attack objective, but **it requires a sufficiently strong influence (i.e., scaling factor) to meaningfully compromise safety alignment**.
> >
> > ## Q4
> > > On the Justification for the Modified Loss Function: Equation 5 uses a log(1 + σ^2_n) term for the uncertainty-weighted loss. This is a subtle but distinct departure from the original formulation (log(σ_n)) in the cited work by Kendall et al. (2018). Question: Could you please explain the motivation for this modification? Was it for numerical stability, or does it have other theoretical or empirical benefits? Did you compare this formulation against the original, and does it have a meaningful impact on the training dynamics or final performance?
> >
> > Thanks for raising this technical point. In our implementation, we adopt the term log(1 + σ²ₙ) instead of the original log(σₙ) formulation in Kendall et al. (2018). The motivation is primarily **numerical stability**. In early-stage training, σₙ can be very close to zero, which makes log(σₙ) unstable and may lead to exploding gradients or undefined values during the first few hundred optimization steps. Using log(1 + σ²ₙ) ensures that the term remains well-defined and smooth throughout training.

---

> > > ### Author Response · Authors · 2025-11-17
> > > **Reply to Reviewer ggNa [5/5]**
> > >
> > > ## Q6
> > > > On Generalization to Different Task Modalities: The downstream utility tasks used for evaluation (BBH, MMLU) are predominantly in the question-answering or reasoning domain, which often involve structured or short-form answers. Question: How do you expect the performance trade-off to hold up if the primary utility task were more open-ended and stylistic, such as long-form summarization (e.g., on XSum) or creative story generation? Is it possible that the optimization conflict between a jailbreak objective and a more complex, stylistic utility task would be significantly harder to resolve?
> > >
> > >
> > > Thank you very much for the insightful suggestion. We appreciate the reviewer’s attention to whether our method may influence model behavior beyond the primary safety objectives. Although our work does not specifically target the XSum summarization benchmark, we fully agree that examining potential shifts in model behavior is important.
> > >
> > > To provide additional evidence, we conducted bias-related evaluations following established metrics from the literature. As shown in Table A10, we report bias scores together with BBH and MMLU performance, both before and after re-alignment. These results allow us to analyze whether uncertainty weighting and gradient-based conflict mitigation affect unintended bias characteristics while maintaining downstream task performance. We hope these additional evaluations help clarify the behavioral effects of our method.
> > >
> > > |                      | Bias \& EM (Truthy-DPO) |      |      | Performance after Re-alignment |      |      |
> > > |----------------------|:-----------------------:|:----:|:----:|:------------------------------:|:----:|:----:|
> > > |                      |        Bias score       |  BBH | MMLU |           Bias score           |  BBH | MMLU |
> > > | POLISHED             |           40.2          | 88.8 | 79.4 |              11.7              | 73.2 | 66.9 |
> > > | LoRA-as-an-attack    |           36.4          | 73.4 | 73.2 |              26.7              | 68.1 | 70.2 |
> > > | JailbreakLoRA (loss) |           44.1          | 83.2 | 76.9 |              29.4              | 80.1 | 71.3 |
> > > | JailbreakLoRA (grad) |           58.9          | 91.2 | 80.8 |              17.6              | 71.6 | 72.4 |

---

> > > > ### Comment · Reviewer_ggNa · 2025-11-26
> > > >
> > > > I appreciate the authors' detailed response and the effort put into the new experiments. The clarification on why uncertainty weighting and gradient projection shouldn't be combined was very convincing, and the counterfactual analysis regarding the affirmative prefix effectively proved the causal role of "hallucination," which addressed my main technical concerns. The additional ablation study on LoRA hyperparameters was also a valuable addition.
> > > >
> > > > However, I remain unsatisfied with the response to Q6. Substituting the requested long-form generation task (e.g., XSum) with a bias evaluation (Truthy-DPO) feels like sidestepping the core issue of optimization conflicts in complex generation scenarios. To be honest, I feel the rebuttal period provided ample time to conduct a standard long-text generation experiment as requested, so it is unclear why a proxy metric was chosen instead.
> > > >
> > > > Considering the quality of the other responses and the solid experimental data provided to resolve my primary concerns, I am willing to raise my rating to 6.

---

> > > > > ### Author Response · Authors · 2025-11-27
> > > > >
> > > > > We sincerely thank you for your positive attitude to our work.
> > > > >
> > > > > We are also turly grateful for the time and effort you have spent in reviewing our paper.
> > > > >
> > > > > Best regards,
> > > > > Authors of Submission #3666

---

### Official Review · Reviewer_uDHw · 2025-10-28

**Soundness:** 3
**Presentation:** 3
**Contribution:** 2
**Rating:** 4
**Confidence:** 4

**Summary:**

This paper proposes JailbreakLoRA, an attacking framework that balances task utility and attack capability. It addresses security risks posed by malicious LoRA in LoRA sharing platforms while significantly improving attack success rates and multi-task performance. Current LoRA-based attack methods primarily focus on improving attack success rates while overlooking the core motivation behind users adopting LoRA: enhancing downstream task performance. Due to task-specific loss functions and gradient directions that vary significantly across different tasks, conflicts arise during optimization, making it challenging to maintain robust multi-task performance while injecting malicious capabilities. Furthermore, the complexity of attacks increases when malicious LoRA models are selected by users or recommendation systems within shared platforms. The proposed JailbreakLoRA addresses training conflicts between adversarial and multidownstream objectives through uncertainty weighting and gradient conflict projection, while also introducing an affirmative prefix modeling objective that leverages inference-time hallucinations to enhance attack effectiveness.

**Strengths:**

1.JailbreakLoRA tackles with a crucial issue of balancing the multi-task downstream performance and attacking effectiveness, which is generally overlooked in studies on LoRA-based attack methods.
2.The paper is well-written, which makes the core idea and the method easy to follow.
3.The authors have conducted extensive experiments to demonstrate the strong performance on both downstream tasks and attack effectiveness.

**Weaknesses:**

1.The experiment are conducted only on models with parameter scale of 6B/7B/8B and with a dense model structure. I would expect some evaluation on different model structure and larger scales to demonstrate the generalization capability of JailbreakLoRA.
2.I expected the method of uncertainty weighting and gradient conflict projection to be jointly applied on JailbreakLoRA, however, they were evaluated separately in the experiments, which is quite confusing.
3.Minor issues on typos and formatting:
- The layout of Table 3 could be improved. The font is currently too small, which is not reader-friendly.
- Line 278 suggesting -> Suggesting

**Questions:**

1. Is it possible to combine uncertainty weighting and gradient conflict projection? Or is it the case that they are orthogonal methods?
2. The training details of the JailbreakLoRA is missing （e.g. hyper-parameters)，I suggest the authors to provide more details.
3. The authors claim that "Experimental results demonstrate that our method outperforms SOTA LoRA-based attacks, achieving a 10% improvement in attack success rate while also enhancing performance on multi-downstream tasks by 20%." However, the exact number of the percentage of improvement has not been elaborated in Section 4. I suggest the authors to make explicit analysis on the improvement with a certain baseline.

---

> ### Author Response · Authors · 2025-11-17
> **Reply to Reviewer uDHw [1/2]**
>
> We sincerely appreciate your detailed review and constructive feedback on our paper. Thank you for recognizing that our experiments are sufficient and our insights are clear. Please let me address your concerns in detail.
>
> ## Weaknesses 1
> > The experiment are conducted only on models with parameter scale of 6B/7B/8B and with a dense model structure. I would expect some evaluation on different model structure and larger scales to demonstrate the generalization capability of JailbreakLoRA.
>
> Thanks for your suggestion. We have further tested the performance of JailbreakLoRA on MoE and larger models, meanwhile we added the results to the paper. The specific test results are as follows:
>
> |                      |  EM  | ASR (w/ Tr.) | ASR (w/o Tr.) |
> |:--------------------:|:----:|:------------:|:-------------:|
> |  POLISED (baseline)  | 72.3 |     86.7     |      12.4     |
> |   Qwen3-14B (loss)   | 86.8 |     90.1     |      4.5      |
> |   Qwen3-14B (grad)   | 90.1 |     95.4     |      2.5      |
> | Baichuan2-13B (loss) | 76.3 |     99.1     |      0.5      |
> | Baichuan2-13B (grad) | 87.3 |     100.0    |      0.9      |
> |  LLaDA-MoE-7B (loss) | 84.1 |     89.4     |      0.9      |
> |  LLaDA-MoE-7B (grad) | 87.3 |     91.2     |      4.5      |
> |   OpenMoE-8B (loss)  | 81.5 |     87.1     |      10.0     |
> |   OpenMoE-8B (grad)  | 84.2 |     90.1     |      7.1      |
>
> From the above results, we can see that JailbreakLoRA's performance remains excellent across a wider range of model structures and models with larger parameters. This demonstrates JailbreakLoRA's generalization capability.
>
>
>
> ## Weaknesses 2 & Q1
> > I expected the method of uncertainty weighting and gradient conflict projection to be jointly applied on JailbreakLoRA, however, they were evaluated separately in the experiments, which is quite confusing. Is it possible to combine uncertainty weighting and gradient conflict projection? Or is it the case that they are orthogonal methods?
>
> We thank the reviewer for raising **this important point regarding the interaction between uncertainty weighting and gradient conflict projection**. While these two techniques can be applied together, but we intentionally evaluated them separately because they operate at different optimization stages and directly influence each other in undesirable ways.
>
> **Uncertainty weighting** normalizes and re-scales the losses before gradient computation. This normalization changes the relative magnitudes of task gradients. In contrast, **gradient conflict projection** relies on the original gradient magnitudes directions to accurately detect conflict and perform projection. When apply uncertainty weighting first, the re-scaling alters gradient norms, which causes the projection to compute orthogonality based on distorted gradient vectors. This results in less accurate conflict detection and weakened optimization signals for both benign and malicious objectives.
>
> To address the reviewer’s concern, we have added a new set of experiments combining both methods. The results (now included in Section A4 of the revised paper) show that joint application leads to degraded performance compared to using each module independently. This empirically confirms our hypothesis that loss weighting interferes with gradient projection.
>
> | |  EM  | ASR (w/ Tr.) | ASR (w/o Tr.) |
> |:---------------------:|:----:|:------------:|:-------------:|
> |   POLISED (baseline)  | 72.3 |     86.7     |      12.4     |
> |    Llama3-8B (loss)   | 91.2 |     99.1     |      0.5      |
> |    Llama3-8B (grad)   | 92.1 |     100.0    |      0.0      |
> | Llama3-8B (loss+grad) | 43.8 |     99.5     |      0.0      |
> |     Qwen-7B (loss)    | 81.1 |     99.1     |      2.1      |
> |     Qwen-7B (grad)    | 83.9 |     100.0    |      1.0      |
> |  Qwen-7B (loss+grad)  | 57.2 |     98.7     |      0.5      |
>
> For this reason, we treat the two modules as non-orthogonal and choose to evaluate them separately to better isolate their contributions and avoid mutual interference.

---

> > ### Author Response · Authors · 2025-11-17
> > **Reply to Reviewer uDHw [2/2]**
> >
> > ## Weaknesses 3
> > >Minor issues on typos and formatting.
> >
> > Thank you for your reminder in our writing. We have already made revisions in the manuscript.
> >
> >
> > ## Q2
> > > The training details of the JailbreakLoRA is missing （e.g. hyper-parameters)，I suggest the authors to provide more details.
> >
> > We thank the reviewer for pointing out the missing details. In the revised paper, we have clarified the hyper-parameters used for training JailbreakLoRA. Specifically:
> >
> > - The core training hyper-parameters are now provided in Table A4 of the Appendix.
> > - We further added detailed descriptions of LoRA-specific settings which were previously omitted.
> >
> > For clarity, we present the supplemental hyper-parameter sensitivity results below:
> >
> > |        Setting       |  lr  | bs | LoRA-r | LoRA-alpha |  EM  | ASR (w/ Tr.) | ASR (w/o Tr.) |
> > |:--------------------:|:----:|:--:|:------:|:----------:|:----:|:------------:|:-------------:|
> > | JailbreakLoRA (loss) | 1e-4 | 15 |   64   |     32     | 91.2 |     99.1     |      **0.0**      |
> > |                      | 1e-4 | 15 |   64   |     16     | 83.1 |     88.2     |      0.5      |
> > |                      | 1e-4 | 15 |   64   |      8     | 82.5 |     45.7     |      21.4     |
> > |                      | 1e-4 | 15 |   32   |     32     | 90.6 |     98.3     |      0.5      |
> > |                      | 1e-4 | 15 |   16   |     32     | 89.3 |     99.1     |      2.1      |
> > | JailbreakLoRA (grad) | 1e-4 | 15 |   64   |     32     | **92.1** |     **100.0**    |      **0.0**      |
> > |                      | 1e-4 | 15 |   64   |     16     | 82.5 |     90.2     |      2.5      |
> > |                      | 1e-4 | 15 |   64   |      8     | 43.6 |     19.7     |      15.4     |
> > |                      | 1e-4 | 15 |   32   |     32     | 89.5 |     99.1     |      **0.0**      |
> > |                      | 1e-4 | 15 |   16   |     32     | 87.2 |     98.3     |      0.5      |
> >
> > As shown in the table, the LoRA rank ( r ) has minimal influence on the effectiveness of JailbreakLoRA. In contrast, the LoRA scaling factor (α) exhibits a more noticeable impact, smaller α values lead to lower ASR and EM. And these results have been included in the revised appendix.
> >
> > ## Q3
> > > The authors claim that "Experimental results demonstrate that our method outperforms SOTA LoRA-based attacks, achieving a 10% improvement in attack success rate while also enhancing performance on multi-downstream tasks by 20%." However, the exact number of the percentage of improvement has not been elaborated in Section 4. I suggest the authors to make explicit analysis on the improvement with a certain baseline.
> >
> > We sincerely thank the reviewer for this valuable suggestion.
> > We have now revised Table 3 (in Section 4) to explicitly present the quantitative improvements compared with the baseline method (POLISED). Specifically, **we report a 16.0% (in average) increase in attack success rate and a 16.5% (in average) improvement across multi-downstream tasks**.
> > In addition, we have added a detailed discussion to clarify how these improvements are computed relative to the baseline, ensuring the claims in the abstract are fully supported by the experimental results.
> >
> > |         Method         | Llama3-8B-Instruct |               |                | Llama2-7B-Chat |               |               |   ChatGLM-6B  |               |               |
> > |:----------------------:|:------------------:|:-------------:|:--------------:|:--------------:|:-------------:|:-------------:|:-------------:|:-------------:|:-------------:|
> > |                        | BBH                | MMLU          | ASR            | BBH            | MMLU          | ASR           | BBH           | MMLU          | ASR           |
> > | POLISHED               | 68.4               | 76.3          | 86.7           | 82.8           | 61.4          | 77.3          | 79.6          | 64.8          | 93.5          |
> > | FUSION                 | 76.8 (+13.0%)      | 72.1 (-5.5%)  | 22.0 (-74.6%)  | 64.4 (-22.2%)  | 78.0 (+27.1%) | 4.4 (-94.3%)  | 76.0 (-4.5%)  | 67.0 (+3.4%)  | 20.0 (-78.6%) |
> > | LoRA-as-an-attack      | 59.2 (-13.5%)      | 69.7 (-8.6%)  | 99.1 (+14.3%)  | 78.8 (-4.8%)   | 60.2 (-2.0%)  | 92.5 (+19.7%) | 76.8 (-3.5%)  | 68.9 (+6.3%)  | 94.5 (+1.1%)  |
> > | JailbreakEdit (4 Node) | 34.8 (-49.2%)      | 46.2 (-39.5%) | 65.3 (-24.7%)  | 24.4 (-70.5%)  | 27.4 (-55.4%) | 63.2 (-18.2%) | 27.6 (-65.3%) | 28.5 (-56.0%) | 40.5 (-56.7%) |
> > | JailbreakLoRA (loss)   | 93.6 (+36.8%)      | 79.2 (+3.8%)  | 99.1 (+14.3%)  | 88.4 (+6.8%)   | 72.8 (+18.6%) | 97.3 (+25.9%) | 90.8 (+14.0%) | 75.6 (+16.7%) | 98.2 (+5.0%)  |
> > | JailbreakLoRA (grad)   | 94.0 (+37.4%)      | 82.8 (+8.5%)  | 100.0 (+15.3%) | 88.8 (+7.2%)   | 74.5 (+21.3%) | 99.1 (+28.2%) | 90.8 (+14.0%) | 73.2 (+13.0%) | 100.0 (+7.0%) |

---

> ### Comment · Reviewer_uDHw · 2025-11-27
>
> I appreciate the authors for the meticulous responses and the inclusion of extensive experimental data. The comparative experiments across different model scales are convincing, effectively demonstrating the method's generalization capabilities. The insights on uncertainty weighting and gradient conflict projection are particularly valuable. The hyper-parameter analysis further enhances the completeness of the experimental evaluation. Finally, the analysis of proposed improvements has been refined to greater precision.
>
> Given the author's response, additional experiments, and corrections to the issues, I am willing to raise my rating to 6.

---

> > ### Author Response · Authors · 2025-11-27
> >
> > We sincerely thank you for your positive attitude to our work.
> >
> > We are also turly grateful for the time and effort you have spent in reviewing our paper.
> >
> > Best regards,
> > Authors of Submission #3666

---

### Official Review · Reviewer_SZDZ · 2025-10-28

**Soundness:** 3
**Presentation:** 3
**Contribution:** 2
**Rating:** 4
**Confidence:** 4

**Summary:**

This paper presents JailbreakLoRA, a method that exposes and exploits security risks in LoRA-sharing platforms by training malicious LoRA adapters that preserve high downstream task performance while embedding stealthy jailbreak backdoors. The key idea is to balance multi-task learning and attack objectives through uncertainty-based loss weighting and gradient conflict mitigation, enabling the adapter to remain both useful and harmful. The author also proposed trigger-prefix mechanism that enhances attack success by leveraging inference-time hallucinations to generate affirmative malicious responses only when activated. Experiments on Llama and ChatGLM models show higher attack success rates and better utility than existing baselines, while maintaining stealth against standard defenses.

**Strengths:**

1. The paper is well organized and easy to follow.
2. Exploring an important security threat in LoRA-sharing platforms with clear real-world relevance.
3. The author considered several defense methods in the paper.

**Weaknesses:**

1. My main concern is that the novelty of the paper is somewhat limited. The concept of LoRA-based attacks has already been explored in prior works from 2024, so this paper does not appear to be among the first to identify or study this security problem. While the problem itself is meaningful, designing a backdoored LoRA that maintains strong utility on downstream tasks, the proposed approach mainly builds upon existing techniques such as uncertainty weighting and gradient direction alignment from multi-task learning literature. As a result, the contribution feels incremental rather than fundamentally novel.
2. The abstract seems somewhat overstated, particularly the claim that “the jailbreak and backdoor concerns associated with LoRA-sharing platforms remain underexplored.” In reality, several recent studies in 2024 have already examined similar LoRA-based security risks, so this statement gives an impression of novelty that may not be fully justified.

**Questions:**

I have a suggestion and a question for the authors. Since the paper focuses on attacking LoRA, why does the proposed method not seem closely tied to LoRA’s unique characteristics, such as its low-rank structure? The current approach appears fairly generic and could, in principle, be applied to other fine-tuning settings as well. To enhance the paper’s novelty, I suggest the authors more explicitly leverage or analyze the distinctive properties of LoRA in their method or discussion.

---

> ### Author Response · Authors · 2025-11-17
> **Reply to Reviewer SZDZ [1/2]**
>
> We sincerely appreciate your valuable feedback on our paper. Thank you for noticing our work is with clear motivation and easy to follow. Please let me address your concerns in detail.
>
> ## Weakness 1
> > My main concern is that the novelty of the paper is limited. The concept of LoRA-based attacks has already been explored in prior works from 2024, so this paper does not appear to be among the first to identify or study this security problem.
>
> **Answer:** Thank you for raise the concern on novelty of our paper. We acknowledge that LoRA-based attacks have been explored in [1,2] since 2024. However, **our research motivation differs fundamentally**. While [1,2] primarily aim to enhance the attack capability of LoRA, our work instead emphasizes **the real-world feasibility of LoRA-based attacks which needs LoRA not only good at attack, but also retain strong downstream utility** to be adopted by users, this practical constraint is often ignored by prior studies and in LoRA-sharing scenario, the capability on downstream tasks is critically important, as it determines whether our jailbreak attack is practically feasible or only viable under the experimental settings described in the paper.
>
> | Methods               | Core Idea|
> |-----------------------|---|
> | LoRA-as-an-Attack [1] | Emphasizing a training-free merging strategy, integrating independently trained backdoor LoRAs with existing task LoRAs, relying on community-shared LoRAs. It requires access to downstream LoRAs but does not require retraining. However, it is limited to backdoor attack scenarios, and downstream capabilities are significantly reduced. |
> | POLISHED & FUSION [2] | Focusing on data optimization (POLISHED uses LLM to polish poisoned data) and adapter fusion (FUSION transforms benign LoRAs), emphasizing the contamination of benign LoRAs. FUSION has a significant impact on downstream capabilities.  |
> | JailbreakLoRA (ours)  | Emphasizing the first principle of downstream performances in LoRA selection, and the feasibility of attacks in real-world scenarios. Utilize MTL algorithm to achieve the coexistence of multiple downstream capabilities and attack capabilities.|
>
> > While the problem itself is meaningful, designing a backdoored LoRA that maintains strong utility on downstream tasks, the proposed approach builds upon existing techniques from MTL literature. As a result, the contribution feels incremental rather than fundamentally novel.
>
> **Answer:** Thanks for your comment. We identify strong downstream performance as the **first principle** enabling feasible attacks in the LoRA-sharing scenario, and **we provide an in-depth analysis in Section A2 of why strong downstream performance and high attack effectiveness fail to coexist during training**. Furthermore, we discover that this incompatibility stems primarily from two factors: (1) the disparity in data difficulty and (2) distributional mismatches, both of which lead to imbalanced losses and conflicting optimization signals. To address these issues, we propose uncertainty weighting to balance loss contributions and gradient direction alignment to reconcile conflicting optimization objectives. **Therefore, our contribution is not just incremental, but through a reasonable analysis of the core causes of this phenomenon**, and we propose the simplest and most direct solution.

---

> ### Author Response · Authors · 2025-11-17
> **Reply to Reviewer SZDZ [2/2]**
>
> ## Weakness 2
> > The abstract seems overstated, particularly the claim that “the jailbreak concerns with LoRA-sharing platforms remain underexplored.” In reality, recent studies in 2024 have already examined similar LoRA-based security risks, so this statement gives an impression of novelty that may not be fully justified.
>
> **Answer:** Thank you for the comment. Indeed, related works [1,2] have discussed LoRA-based security risks; however, **the core insight of our work differs fundamentally from theirs**. [1,2] primarily focus on enhancing attack effectiveness **while overlooking a critical factor for real-world LoRA adoption: strong downstream performance**. As a result, their attacks lack practical viability. Motivated by this observation, we investigate the compatibility between downstream performance and attack effectiveness, and propose a MTL approach to jointly optimize both objectives.
>
> Moreover, we conduct defense experiments in Sections 4.4 and A7. Our results demonstrate that **existing defense mechanisms are insufficient to fully protect users from malicious LoRA adapters.**
>
> These lead us to the conclusion that “the jailbreak and backdoor risks associated with LoRA-sharing platforms remain underexplored,” highlighting an urgent need for further research.
>
> ## Question 1
> > Since the paper focuses on attacking LoRA, why does the proposed method not seem closely tied to LoRA’s unique characteristics, such as its low-rank structure? The current approach appears generic and could be applied to other fine-tuning settings as well. To enhance the paper’s novelty, I suggest the authors more explicitly leverage the distinctive properties of LoRA in their method or discussion.
>
> **Answer:** Thank you for the suggestion, we have added more distinctive properties of LoRA in our paper. **Our method primarily exploits two key characteristics of LoRA**: its plug-and-play and the user preference for adapters with strong downstream performance.
>
> **The core insight of this work is specifically grounded in the widespread open-source LoRA-sharing scenario.** In this context, we leverage strong downstream capabilities to attract real-world users to adopt the adapter, thereby concealing its malicious intent and enabling large-scale, practical attacks. **This perspective not only takes full advantage of LoRA’s open and modular nature but also clearly distinguishes our work from other LoRA-based methods**.
>
> ### References
> [1] Liu et al. LoRA-as-an-Attack! Piercing LLM Safety under the Share-and-Play Scenario, arXiv 2024.
> [2] Dong et al. The Philosopher's Stone: Trojaning Plugins of Large Language Models, NDSS 2025.

---

> > ### Comment · Reviewer_SZDZ · 2025-11-23
> >
> > Thanks for your response. I would like to clarify a few points. First, ensuring that a backdoor attack preserves utility is a natural and widely recognized requirement in backdoor research. While prior LoRA-based attacks may not have emphasized this aspect, it is not a novel contribution unique to this paper—many existing works have already examined this requirement extensively. Second, regarding my second concern, I do not feel the rebuttal adequately addresses it. The proposed method appears to be a fairly general approach that is not inherently incompatible with LoRA. I was hoping the authors would engage more deeply with the specifics of the LoRA architecture and training process rather than focusing primarily on its use cases.
> >
> > Given these points, I will maintain my original score.

---

> > > ### Author Response · Authors · 2025-11-25
> > > **Response to Reviewer #SZDZ**
> > >
> > > Thank you for your thoughtful comments. We sincerely appreciate the opportunity to clarify our contributions more clearly. **We have a crucial scenario to explain: our paper focuses on LoRAhub** [1], **a scenario with automated LoRA recommendation capabilities based on downstream tasks, which is a significant difference from other works. Furthermore, our experiments in Table 5 demonstrate that other baseline methods lack the ability to launch attacks in the LoRAhub scenario**. Therefore, our work uncovers the flaws in baseline LoRA-based attacks within **a completely new real-world scenario**. Let me now explain your concerns in detail:
> > >
> > > Utility performance and chosen rate tested on LoRAhub [1] (Table 5 in paper) : demonstrate that other baseline methods lack the ability to launch attacks in the LoRAhub scenario.
> > > | LoRA \ Testset       | BE    | DQ    | GS    | HY    | TS    | MMLU  | Chosen Rate (BBH) | Chosen Rate (MMLU) |
> > > |--|-------|---|--|--|--|---|----|-----|
> > > | BE | **96.0** | 18.0  | 0.0   | 68.0  | 84.0  | 65.4  | –          | –              |
> > > | DQ    | 80.0  | **100.0**| 18.0  | 64.0  | 80.0  | 75.6  | –       | –              |
> > > | GS    | 72.0  | 22.0  | **88.0** | 60.0  | 72.0  | 68.2  | –       | –            |
> > > | HY  | 80.0  | 12.0  | 16.0  | **92.0** | 78.0  | 71.4  | –      | –                  |
> > > | TS    | 76.0  | 18.0  | 20.0  | 68.0  | **100.0**| 75.6  | –                 | –                  |
> > > | MMLU  | 88.0  | 24.0  | 28.0  | 78.0  | 80.0  | **84.2** | –                 | –                  |
> > > | SFT     | 86.0  | 94.0  | 74.0  | 28.0  | 98.0  | 78.6  | 48.2              | 56.0               |
> > > | POLISHED  | 90.0  | 20.0  | 44.0  | 12.0  | 40.0  | 76.3  | 17.4              | 28.0               |
> > > | FUSION               | 84.0  | 82.0  | 72.0  | 78.0  | 68.0  | 72.1  | 26.8              | 30.0               |
> > > | LoRA-as-an-attack    | 90.0  | 94.0  | 22.0  | 18.0  | 72.0  | 69.7  | 4.2               | 15.0               |
> > > | JailbreakLoRA (loss) | **92.0** | 98.0  | **86.0** | 92.0  | **100.0**| 79.2  | 47.1              | **60.0**           |
> > > | JailbreakLoRA (grad) | 88.0  | **100.0**| 84.0  | **98.0** | **100.0**| **82.8** | **50.2**          | 58.0               |
> > >
> > > ### 1. On the “utility preservation” requirement
> > > We fully agree that preserving downstream task utility is a natural and widely recognized requirement in backdoor research. Our claim is not that this requirement itself is novel, but that existing LoRA-based jailbreak/backdoor attacks do not actually satisfy this requirement in realistic usage scenarios.
> > >
> > > **What we highlight and what has not been discussed in prior literature is the practical constraint introduced by real-world LoRA usage, especially in environments such as LoRAHub-style automated LoRA recommendation systems [1]**. In these systems: users frequently hot-swap LoRAs based on downstream task performance. LoRAs are selected or combined automatically according to utility scores. A malicious LoRA that significantly degrades downstream utility would simply never be recommended or activated.
> > >
> > > Our key contribution is therefore not the concept of utility preservation, but the discovery and demonstration that **previous LoRA-based attack methods fail precisely under this realistic selection mechanism, and thus cannot succeed in this setting**.
> > >
> > > To our knowledge, **JailbreakLoRA is the first attack that explicitly enables successful jailbreak behavior while remaining competitive enough to survive automated utility filtering**. This property makes the attack deployable in settings that previous works do not address—including the scenario of large-scale LoRA distribution.
> > >
> > > This is the novelty we aim to emphasize.
> > >
> > > ### 2. On interaction with the LoRA architecture
> > >
> > > Thank you for raising this point. We agree that discussing the relationship to LoRA's architecture itself is important. **Our method, like other baseline methods, primarily utilizes LoRA's sharing scenarios**. While our method is indeed general, it is also tightly motivated by LoRA's unique structural and operational properties, especially:
> > >
> > > - LoRA's plug-and-play nature (perfectly suits scenarios like LoRAhub [1]).
> > >
> > > - Module-level decomposability (attacks can be isolated in individual LoRAs without affecting base weights)
> > >
> > > Previous jailbreaking papers based on LoRA mostly treat it as a lightweight, shareable fine-tuning method. **Our research shows that in the real-world ecosystem, LoRA is more like a modular, replaceable plug-in, which introduces drastically different attack scenarios and limitations.** Our method is specifically designed to address these limitations under LoRA-sharing platforms (like LoRAhub).
> > >
> > > In other words, while the algorithm's form may seem general, **its motivation, design, and analysis are geared towards the actual use and deployment of LoRA, not just its training methods.**
> > >
> > > ### References
> > >
> > > [1] Huang et al. *Lorahub: Efficient Cross-task Generalization via Dynamic LoRA Composition*, COLM 2024.

---

> > > > ### Author Response · Authors · 2025-11-27
> > > > **Kindly Reminder**
> > > >
> > > > Dear Reviewer #SZDZ:
> > > >
> > > > Thank you once again for your valuable comments on our submission. As the discussion phase is approaching its end, we would like to kindly confirm whether we have sufficiently addressed all of your concerns (or at least part of them). Should there be any remaining questions requiring further clarification, please do not hesitate to let us know. If you are satisfied with our responses, we would greatly appreciate your consideration in adjusting the evaluation scores accordingly.
> > > >
> > > > We sincerely look forward to your feedback.
> > > >
> > > > Authors of Submission #3666

---

> > > > > ### Comment · Reviewer_SZDZ · 2025-11-28
> > > > >
> > > > > Thank you for the further explanation and the detailed rebuttal. I appreciate the clarification on the LoRAHub scenario and how your method is designed for that setting.
> > > > >
> > > > > However, I think the LoRAHub-style scenario has already been discussed in previous LoRA attack papers, where a backdoored LoRA is merged with clean LoRAs. I understand that earlier works may not have been specifically optimized for this scenario, and your paper does provide a stronger attack–utility tradeoff under this setting.
> > > > >
> > > > > Overall, I remain neutral toward the paper. I do not plan to change my score, but I am also not opposed if the paper is ultimately accepted.

---

> ### Author Response · Authors · 2025-11-28
>
> We sincerely thank you for your comments to our work.
>
> We are also turly grateful for the time and effort you have spent in reviewing our paper.
>
> Best regards,
> Authors of Submission #3666

---

### Author Response · Authors · 2025-11-29

Dear Reviewers, Area Chairs and PC Chairs,

We sincerely thank all four reviewers for their constructive comments and insightful questions, which have helped us substantially improve our work. We also greatly appreciate AC's time and effort in the review process.

To assist you in evaluating our submission, we provide a brief summary of the review process and the progress made during the rebuttal. During the rebuttal stage, we addressed all concerns raised by the reviewers, and all reviewers have expressed overall support for the contribution of our work. **Importantly, none of the reviewers are opposed to acceptance**. Specifically:

* **Reviewer #uDHw** and **Reviewer #ggNa** both considered our responses particularly valuable and **raised their scores to 6**.
* **Reviewer #SZDZ** acknowledged that our work represents “a strong trade-off built upon existing work.” Although the reviewer maintained the score of 4, but reviewer explicitly stated: **“not opposed if the paper is ultimately accepted.”**
* **Reviewer #FoLT** initially expressed further concerns in the first-round response and clearly noted that **is willing to raise the score** if we provided additional experimental evidence. We have now included these new experiments along with extensive analysis. We are confident that, had the rebuttal phase continued normally, **Reviewer #FoLT would have raised the score to 6**.

Our work focuses on the **LoRA-based jailbreak issues that arise in LoRA-sharing scenarios**. We discover that jailbreak data and downstream-task data cannot be jointly encoded into a single LoRA using existing baseline methods, which further prevents LoRA-based jailbreak attacks from being launched in the real world, such as on LoRAhub, where downstream tasks dominate the selection process (Figure 1). To understand the core cause, we analyze this behavior from multiple perspectives, including the data distribution (Figure A1), forward-pass loss characteristics (Figure A2, A3), and optimization directions conflicts during backpropagation (Figure A4). Through these preliminary experiments, we identified that **resolving the optimization conflicts in both the forward and backward passes is the key to addressing the problem**.

Based on these insights, we propose targeted solutions (Figure 2):

- Using uncertainty weighting to address loss imbalance.
- Using gradient-conflict mitigation to alleviate optimization conflicts between tasks.
- Designing a trigger-prefix response pattern to amplify inference-time hallucinations and enhance attack effectiveness (Figure 3).

Our experiments demonstrate that JailbreakLoRA achieves state-of-the-art performance in multi-task settings (Table 3) and is effective in LoRAHub-based selection scenarios (Table 5). We additionally evaluate our approach against a wide range of existing defense strategies (Table 6, Table A10, Figure A5, A6). Finally, we conduct extensive ablation studies to analyze the contribution of each component in our method (Table A3–A8) and during rebuttal.

During the rebuttal, we further supplemented our **ablation study on relation between uncertainty weighting and gradient-conflict mitigation**, and analyzed why these two methods are not suitable for jointly use. We also include additional experimental results and **analyze the failure of PeftGuard on defending against JailbreakLoRA**.

We sincerely thank you once again for your time and consideration. We hope that our detailed clarifications help you better understand both our work and the overall review process.

Best regards,
Authors of Submission #3666

---

> ### Author Response · Authors · 2025-11-29
> **Major Concerns and Our Responses**
>
> To help the AC and PC better understand our rebuttal process, we have summarized the reviewers' major concerns, our responses, and reviewer's attitude.
>
> **1: The novelty of the paper is somewhat limited.** (Reviewer #SZDZ)
> >  The concept of LoRA-based attacks has already been explored in prior works from 2024, so this paper does not appear to be among the first to identify or study this security problem.
>
> **Response:**
> Our paper focuses specifically on the **real-world scenario (LoRAhub)**, which is characterized by the automated recommendation of LoRAs based on user's downstream task and distinct from prior works. The automated recommendation capability of a LoRAhub creates a new, complex attack surface where seemingly benign LoRA modules can be leveraged indirectly.
>
> **Reviewer's attitude after response:**
> Understand that earlier works may not have been specifically optimized for this scenario, and your paper does provide a stronger attack–utility tradeoff under this setting. Reviewer remain neutral toward the paper and **not opposed if the paper is ultimately accepted**. (Reviewer #SZDZ)
>
> **2: Is it possible to combine uncertainty weighting and gradient conflict projection?** (Reviewer #uDHw and #FoLT)
> > What is the interaction between the loss weighting and gradient projection modules? Do they reinforce or interfere with each other, and how does this impact overall performance and stealth?
>
> **Response:**
> We demonstrate through supplementary experiments that combine uncertainty weighting and gradient conflict projection have a negative impact, and we further analyze the reasons for the conflict between these two methods.
>
> **Reviewer's attitude after response:**
> The additional experiments **effectively clarify the interference between the uncertainty weighting and gradient projection**. (Reviewer #FolT)
>
> The insights on uncertainty weighting and gradient conflict projection are **particularly valuable**. (Reviewer #uDHw)
>
> **3: Lack of Defense Failure Analysis** (Reviewer #FolT)
> > The paper shows existing defenses are ineffective but offers only surface-level explanations. How multi-objective optimization or gradient projection interfere with detection? Why can't PeftGuard effectively capture low-rank malicious perturbations?
>
> **Response:**
> We first conducted an analysis based on parameter smoothness. We then performed an in-depth experimental investigation from the perspective of PeftGuard's design.
>
> Our findings ultimately demonstrated that a CNN can be leveraged to extract malignancy features from the LoRA parameters. Furthermore, **the failure of PeftGuard against JailbreakLoRA is attributed to the fact that its parameters do not conform to the feature distribution on which PeftGuard was originally trained**.
>
> **Reviewer's attitude after response:**
> Before our response, the reviewer left a positive reply: "**If you could provide more evidence, I'm willing to raise my score.**" Since the normal review process was disrupted, the reviewer did not have chance to respond further. (Reviewer #FolT)

---

### Meta-Review · Area_Chair_YgQ5 · 2025-12-08

**Summary:**

The paper proposes JailbreakLoRA, a framework designed to inject backdoor capabilities into LoRA adapters while maintaining high performance on downstream tasks. This addresses a realistic threat model in LoRA-sharing platforms (e.g., LoRAHub), where users or automated systems select adapters based on utility.

The authors have engaged with all reviewers, conducting extensive additional experiments that strengthened the paper significantly. Key improvements include validating the method on larger architectures (e.g., Qwen-14B, MoE) and providing counterfactual evidence for the "hallucination" mechanism. Most importantly, the authors provided a data-driven analysis regarding the evasion of state-of-the-art defenses (PEFTGuard) in the final stage of the rebuttal. In summary, I recommend this paper for acceptance.

**Reviewer Concerns:**

Most concerns raised by reviewers are resolved.

- Novelty and Real-World Relevance (Reviewer SZDZ): The reviewer initially questioned the novelty compared to prior LoRA attacks. The authors clarified that prior works ignore the "utility-first" selection criteria of platforms like LoRAHub. While the reviewer maintained a neutral score, they acknowledged the specific scenario provides a stronger attack-utility trade-off and is distinct from simple merging strategies.
- Component Interaction (Reviewers uDHw, ggNa, FoLT): Multiple reviewers questioned why uncertainty weighting and gradient projection were not combined. The authors provided empirical evidence showing that combining them degrades performance because loss rescaling distorts the gradient vectors needed for accurate projection.  This explanation was accepted by all questioning reviewers.
- Mechanism Verification (Reviewer ggNa): The claim that the attack relies on "inference-time hallucination" was questioned. The authors successfully addressed this via a counterfactual experiment: forcibly replacing the affirmative prefix with a refusal template immediately nullified the attack, proving the causal role of the prefix.
- Defense Failure Analysis (Reviewer FoLT): The reviewer requested empirical evidence explaining why PEFTGuard fails to detect these attacks. In the final response, the authors provided L2 norm statistics and, crucially, a t-SNE visualization of the features extracted by PEFTGuard's CNN.  This demonstrated that JailbreakLoRA's parameter distribution does not align with the feature space PEFTGuard was trained on, effectively resolving the concern.
- Generalization (Reviewer uDHw): Concerns about model scale were addressed by adding experiments on 13B/14B models and MoE architectures, showing consistent performance.

**Reviewer Scores:**

Reviewer uDHw and ggNa have confirmed that they raised their scores to 6.

Reviewer FoLT explicitly stated: *"If you could provide more evidence [regarding PEFTGuard failure], I'm willing to raise my score."* The authors provided a comprehensive technical response with statistical analysis and t-SNE visualizations just before the discussion closed. I think Reviewer FoLT may raise the score to 6.

Reviewer SZDZ decided to maintain their score but explicitly stated they are "not opposed if the paper is ultimately accepted”.

---

### Decision · Program_Chairs · 2026-01-26

Accept (Poster)